# Incorporation and Deposition of Spin Crossover Materials into and onto Electrospun Nanofibers

**DOI:** 10.3390/polym15102365

**Published:** 2023-05-18

**Authors:** Maximilian Seydi Kilic, Jules Brehme, Justus Pawlak, Kevin Tran, Friedrich Wilhelm Bauer, Takuya Shiga, Taisei Suzuki, Masayuki Nihei, Ralf Franz Sindelar, Franz Renz

**Affiliations:** 1Institute of inorganic Chemistry, Leibniz Universität Hannover, Callinstraße 7, 30167 Hannover, Germany; 2Faculty II, Hochschule Hannover, University of Applied Science an Arts, Ricklinger Stadtweg 120, 30459 Hannover, Germany; 3Hannover School for Nanotechnology, Laboratorium für Nano-und Quantenengineering (LNQE), Leibniz Universität Hannover, Schneiderberg 39, 30167 Hannover, Germany; 4Graduate School of Pure and Applied Sciences, University of Tsukuba, Tennodai 1-1-1, Tsukuba 305-8577, Ibaraki, Japan

**Keywords:** spin crossover, (coaxial)-electrospinning, triazole complexes, nano fibers, PMMA, coordination chemistry, composites

## Abstract

We synthesized iron(II)-triazole spin crossover compounds of the type [Fe(atrz)_3_]X_2_ and incorporated and deposited them on electrospun polymer nanofibers. For this, we used two separate electrospinning methods with the goal of obtaining polymer complex composites with intact switching properties. In view of possible applications, we chose iron(II)-triazole-complexes that are known to exhibit spin crossover close to ambient temperature. Therefore, we used the complexes [Fe(atrz)_3_]Cl_2_ and [Fe(atrz)_3_](2ns)_2_ (2ns = 2-Naphthalenesulfonate) and deposited those on fibers of polymethylmethacrylate (PMMA) and incorporated them into core–shell-like PMMA fiber structures. These core–shell structures showed to be inert to outer environmental influences, such as droplets of water, which we purposely cast on the fiber structure, and it did not rinse away the used complex. We analyzed both the complexes and the composites with IR-, UV/Vis, Mössbauer spectroscopy, SQUID magnetometry, as well as SEM and EDX imaging. The analysis via UV/Vis spectroscopy, Mössbauer spectroscopy, and temperature-dependent magnetic measurements with the SQUID magnetometer showed that the spin crossover properties were maintained and were not changed after the electrospinning processes.

## 1. Introduction

The reversible transition between high-spin (HS) and low-spin (LS) states can occur in the octahedral ligand field of metal coordination compounds of elements with electron configurations from 3 d^4^ to 3 d^7^. This transition is known as spin crossover and is triggered by external stimuli such as temperature, pressure, guest molecules, or chemical influences and irradiation [1,2]. Spin crossover can be seen as a paradigm for the bistability of spin states at a molecular level [3,4]. The spin crossover (SCO) phenomenon is accompanied by various property changes. Compounds containing Fe^II^, for example, show a switch between para- and diamagnetic behavior [5]. The spin state change is also accompanied by a change of the optical properties, electronic properties, and a change of the metal to ligand bond length [6]. Therefore, spin crossover materials represent a very active field of research. Through the change of properties, they could possibly find applications within mechanical, electronic, photonic, and optical devices [7,8].

Complexes with Fe^II^ and triazole ligands are known to have suitable ligand field strengths so that SCO effects can be observed. This is also possible for these complexes around ambient temperature, and below as well as above [9,10]. When triazole ligands are used, which are substituted at the 4-position, the ligands and central atoms form one-dimensional coordination chains. Their general form is described by the formula [Fe(Rtrz)_3_]X_2_, with Rtrz being a 4-substituted-1,2,4-triazole ligand, and X being a monovalent anion like the chloride- or 2ns-anion (Figure 1) [8,11]. The SCO properties of these coordination chains can be influenced by changing the counterions or changing the substitution of the 4-position. We chose triazole complexes because the modification of their SCO properties is possible by simply changing the educts during the synthesis and because they exhibit high chemical stability. In this context, it is also possible that different ligands are combined or counterions are used in various ratios to influence the resulting SCO properties of the complex [8,12]. If a change of the SCO properties is desired, post-synthetic modification could also be performed [13].

The high chemical stability is triggered by the triple N^1^, N^2^-triazole bridges connecting the metal centers in the coordination chains. The reason for the increased chemical stability lies in the bond angles between the metal atoms and the nitrogen of the ligands. These correspond approximately to the exocyclic donor electrons of the five-membered ring, which means that the emerging ring strain is low [8,10]. The high chemical stability is particularly advantageous with regard to the electrospinning process we have been using. In the electrospinning process, the complexes used are subjected to harsh conditions, which can lead to the oxidation of the iron, which happened in previous studies [14]. The change of the oxidation state can lead to a less cooperative interaction of the remaining iron(II)-centers, and therefore impact the SCO behavior, resulting in more difficulties in controlling the desired properties of the composite. If there is no change or only a slight change in the SCO properties after implementation, no further chemical modification of the complexes is necessary, and the composite production can be simplified. Generally, the SCO behavior of the composites is not completely predictable. It is common for the SCO behavior to vary depending on the used polymer and its concentration. This has an influence on the switching temperature and the hysteresis of the composite material with the abruptness of the process being decreased in previous cases, as the thermal conductivity of the composite is lower compared to the pure complex [15,16,17]. For example, in attempts where an SCO PMMA composite blend was obtained, the resulting switching temperature and hysteresis of the composite showed a significant difference in comparison to the complex itself [17].

Electrospinning is a simple technique, with a potential to scale up and become a versatile production technology to produce nanofibers out of polymer solution. Although the typical fibers in this process exceed the 100 nm range, they are still called nanofibers in the aspect of engineering and in the industry [18]. In a typical electrospinning process, an electrical potential (usually between 5 and 30 kV) is applied between a droplet of the solution in a syringe needle and a grounded collector. As soon as the Coulomb forces acting on the droplet, due to the applied voltage, overcome the surface tension of the polymer solution, a fiber jet emerges from the apex of the conical meniscus known as the Taylor cone [19,20]. As the liquid fiber dries, an electrically charged fiber remains, which discharges on the collector. In our case, we use a rotating collector to obtain aligned fibers depending on the velocity. Starting in 1979, experiments without needles (needleless electrospinning NLES) started to emerge, as the polymeric jet is a self-organized process and is not prompted by capillary forces. However, it comes with a harder-to-control process [21,22].

Coaxial electrospinning was first introduced in 2002, enabling a wider application range for nanofibers and diversifying their morphology [23]. Non-polymeric materials (ceramics, metal oxides, or semiconducting materials) could especially be used in the electrospinning process. However, it comes at the price of an increased number of parameters to be taken care of, including the shell-to-core fluid flowrate miscibility and compatibility. Since then, the coaxial method has been the base for more multichannel electrospinning systems (e.g., triaxial, tetra-axial) [24,25,26]. It is important to note that the viscosity of the shell solution has to be higher than that of the core solution, as the requirement to form stable core–shell fibers is to overcome the interfacial surface tension between the core and shell solution. Additionally, the flowrate of the shell solution needs to be higher than the core solution so that the core remains completely entrained [27].

A PMMA-based system for the application in a polymer-based waveguide was used before by a cooperative working group, so the combination of this class of complexes (triazole complexes) and PMMA was proven to be functional [28]. Using electrospinning as the fabrication method could potentially lead to smaller and more efficient waveguides. Through the controlled implementation of SCO particles into optically active polymer-based fibers, a simple and targeted addressing of SCO compounds could be possible. We are now able to determine the position of the complex particles inside of the complex polymer composite by using SEM pictures and EDX measurements. This was performed to gain further information about this type of composite fiber and further evaluate its potential for polymer-based waveguides. Subsequently, we used coaxial electrospinning to add a protective shell to the resulting polymer fibers for one attempt so that the attached complexes could not be detached from the fibers via mechanical action or water contact. Therefore, we report that the coaxial electrospinning approach resulted in core–shell-like structures with the composite as the core material and an additional shell of PMMA.

## 2. Materials and Methods

### 2.1. General

The used Fe^II^-triazole complexes and composite materials were synthesized by using the following purchased chemicals without further purifying them: Iron(II)Chloride tetrahydrate (FeCl_2_ · 4 H_2_O) (>99%) from Sigma-Aldrich (St. Louis, MO, USA); L-ascorbic acid (>99%) from Carl Roth; 4-Amino-1,2,4-Triazole (99%) purchased from Thermo Scientific (Waltham, MA, USA); Sodium 2-Naphthalenesulfonate (98%) from Alfa Aesar (Haverhill, MA, USA); PMMA 350,000 Mw from Sigma-Aldrich (St. Louis, MO, USA); and 2,2,2-Trifluorethanol (TFE) from Carl Roth. The measured Mössbauer spectra were recorded in transmission, and ^57^Co/Rh source was used.

Two different complexes were used in the two approaches performed, with only the corresponding anions being changed. This change in anions should result in slight differences in the thermally induced SCO effect, but it is not expected to significantly affect other properties of the composite material.

The decision regarding the applied potential and concentration for the spinning parameters was previously made by our working group, as these values were found to yield the best results in this context.

### 2.2. Synthesis of [Fe(atrz)_3_]Cl_2_

A modified synthesis that was based on a synthesis by Bousseksou et al. [29] was performed. Therefore, 0.5887 g of FeCl_2_ · 4 H_2_O was dissolved in 1.25 mL H_2_O with 20 mg of L-ascorbic acid. Separately, 0.746 g of 4-Amino-1,2,4-Triazole was dissolved in 1.25 mL H_2_O. The iron solution was added to the triazole solution, and the resulting solution was stirred for 2 h. A white solid was obtained in the process. The solid was then further purified by dispersing it in ethanol, followed by centrifugation at 6000 rpm for 10 min for 3 times. In the process, the solid changed color from white to pink. The product was subsequently dried in the air and 1.08 g was obtained (Yield: 85%).

After drying, average particle diameters of 2.37 µm of the then agglomerated particles could be determined via SEM.

Analytically found (calculated) with CHN elemental analysis for C_6_H_12_N_12_Cl_2_·2.85H_2_O (molar mass 430.33 g mol^−1^): C, 16.91 (16.75); H, 3.47 (4.15); N, 38.98 (39.06). Far-infrared (FIR) (in cm^−1^): 469 (w), 479 (w), 515 (w). Mid-infrared (MIR) (in cm^−1^): 623 (s); 701 (s); 852 (w); 869 (m); 891 (m); 1001 (m); 1031 (w); 1063 (m); 1100 (s); 1219 (s); 1313 (w); 1357 (w); 1401 (w); 1486 (w); 1543 (m); 1618 (s); 1663 (s); 3014 (w); 3081 (m); 3113 (s); 3197 (m); 3265 (s); 3301 (s); 3400 (s).

### 2.3. Synthesis of [Fe(atrz)_3_](2ns)_2_

First, the corresponding iron(II) salt had to be obtained based on a synthesis by Caseri et al. [30]; then, the complex was also synthesized following the modified synthesis by Bousseksou et al. [29] Therefore, to obtain Fe(2ns)_2_ · 6 H_2_O, 2.5 g of Sodium 2-Naphthalenesulfonate was dissolved in 75 mL of H_2_O by heating up to 70 °C and stirring at 650 rpm. A dull solution was obtained. Separately, 1.08 g of FeCl_2_ · 4 H_2_O was dissolved in 2.5 mL of H_2_O and then added to the Sodium 2-Naphthalenesulfonate solution. A white solid precipitated from the solution, which was then washed 3 times with 150 mL of water. The white solid was then dried in a desiccator under vacuum, and 1.9978 g (yield: 3.45 mmol, 64%) was obtained. The white precipitate was further analyzed via IR spectroscopy to confirm that Fe(2ns)_2_ · 6 H_2_O was obtained. MIR (in cm^−1^): 612 (m); 621 (m); 645 (m); 668 (m); 736 (w, broad); 758 (s); 815 (s); 906 (w); 943 (w); 964 (w); 1033 (s); 1091 (m); 1181 (s, broad); 1347 (w); 1503 (m); 1592 (m); 1646 (s); 1670 (w); 1981 (w); 2364 (w, broad); 3061 (w); 3364 (s, broad).

Then, 0.8562 g of the obtained iron(II) salt was dissolved in 4 mL of methanol. Separately, 0.373 g of 4-Amino-1,2,4-Triazole was dissolved in 3 mL of H_2_O. The solution of the iron(ii) salt was then added to the 4-Amino-1,2,4-Triazole solution and was stirred for 2 h. Thereby, a pink precipitate was formed. The obtained solid was then purified by dispersing it in ethanol and centrifuging it 3 times at 6000 rpm for 10 min. The obtained solid was then dried in a desiccator, and 0.632 g was obtained (Yield: 0.8245 mmol, 56%).

After drying, average particle diameters of 2.48 µm of the then agglomerated particles could be determined via SEM.

Analytically found (calculated) with CHN elemental analysis for C_26_H_26_N_12_O_6_S_2_·2.44H_2_O (molar mass 766.49 g mol^−1^): C, 40.55 (40.74); H, 3.67 (4.06); N, 21.77 (21.93). FIR (in cm^−1^): 474 (m); 502 (m); 552 (s); 560 (s); 568 (s); 622 (s); 647 (m); 675 (s); 748 (s); 768 (w); 819 (s); 865 (s); 906 (s); 944 (m); 956 (m); 981 (w); 1032 (s); 1063 (m); 1093 (s); 1138 (s); 1184 (s); 1271 (s); 1346 (w); 1383 (w); 1446 (m); 1504 (w); 1544 (w); 1593 (w); 1628 (m, broad); 3011 (w); 3060 (m); 3073 (w); 3134 (w); 3163 (m); 3214 (w); 3283 (m, broad); 3498 (m, broad).

### 2.4. Electrospinning of PMMA Fibers with [Fe(atrz)_3_]Cl_2_

A solution was prepared dispersing 0.227 g [Fe(atrz)_3_]Cl_2_ in 10 mL TFE (2,2,2-Trifluoroethanol) and sonicating the solution for 1 h. Then, 1.35 g PMMA was added to the solution and it was stirred overnight for 12 h to obtain a homogeneous solution. The mixture was electrospun at 18 kV with a pumping rate of 1 mL/h, with a collector speed of 10 m/s, a needle diameter of 0.8 mm, and a needle collector distance of 20 cm at room temperature, with an air humidity of 42%, and the polymer complex composite (PCC) was collected on an aluminum foil, as schematically shown in Figure 2.

### 2.5. Coaxial Spinning of PMMA Fibers with [Fe(atrz)_3_](2ns)_2_

For the shell solution, 1.5 g PMMA was dissolved in 10 mL TFE and stirred overnight for 12 h to obtain a homogenous solution. The core solution contained 500 mg of [Fe(atrz)_3_](2ns)_2_ which was dispersed in 10 mL TFE and sonicated for 1 h to gain uniformly sized particles before 0.5 g PMMA was added to the solution. Following that, the solution was stirred overnight for 12 h to obtain a homogeneous solution. Those mixtures were electrospun at 18 kV, 1.3 mL/h for the shell and 0.9 mL/h for the core with a collector speed of 10 m/s, an outer needle diameter of 0.8 mm (inner diameter of 0.514 mm, 21 Gauge), an inner needle diameter of 0.33 mm (inner diameter of 0.184 mm, 29 Gauge), and a needle collector distance of 20 cm at room temperature, with an air humidity of 42%, while the fibers were collected on an aluminum foil, as pictured in Figure 3.

### 2.6. Characterization

Infrared spectroscopy was performed to gain information about the molecular structure of the complex, polymer, and composite. For this, a Perkin Elmer spectrum two was used with the ATR method between 500 and 3000 cm^−1^.

The UV/Vis was performed using a lambda 650 S from Perkin Elmer from 800 to 250 nm in 1 nm steps. The fiber mats were placed in the reflectance sample holder of the 150 mm integrating sphere with a heater behind it to increase the temperature of the sample up to the spin transition, to determine if the spin transition still occurs.

For the SEM images, a Carl Zeiss Supra 55 VP was used. For the EDX measurement, an Oxford XMax 80 mm^2^ was used. The EDX was used to obtain information about the place of the complex in the fiber structure.

The Mössbauer measurements were carried out in transmission geometry with a modified miniaturized Mössbauer Spectrometer MIMOS II (Space and Earth Science Instrumentation) at room temperature. The gamma radiation source was ^57^Co nuclei in an Rh matrix. The measurements were recorded with 14.4 keV and all isomer shifts were given relative to α-Fe.

Direct current magnetic susceptibility measurements of powder samples and those in PMMA were measured with a Quantum Design MPMS-5XL SQUID magnetometer under an applied magnetic field of 1000 Oe. Each sample was wrapped in an Al foil and fixed in an Al cup. The temperature dependence was measured at 2 K increments with a sweeping rate of 2 K/min. Data were corrected for the paramagnetic contribution of the sample holder and the diamagnetic contribution of the sample calculated from Pascal’s constants, except for PMMA.

CHN elemental analyses were performed using Perkin-Elmer 2400 II CHN analyzer.

## 3. Results

### 3.1. Electrospun PMMA Nanofibers with [Fe(atrz)_3_]Cl_2_

Through the uniaxial electrospinning process, we obtained a pink-colored fiber mat. In Figure 4a, the IR-Spectrum of the composite product in which the complex shall be deposited on the outer surface of the fibers is shown. This is shown in comparison to the IR-spectra of the pure PMMA nanofibers and the pure complex. The C=O band at 1750 cm^−1^ is clearly assignable to the PMMA, as there are none in the ligands of the complex. Through the comparison, it can be seen that some of the most important bands of both the pure PMMA nanofibers and the complex are still visible in the composite product. The bands also show no shift in their wavenumber, indicating no chemical interaction between the complex and the nanofibers. This leads us to the assumption that the complex was still present after the electrospinning process and did not succumb to the harsh conditions of the spinning process. Due to the ratio of complex to polymer, the bonds of the polymer are overlapping the bonds of the complex; nevertheless, it can be seen in the fingerprint area that the triazole bonds are still visible and cannot be assigned otherwise to the polymer.

As further proof that the complex is inside the composite product, a Mössbauer spectrum at ambient temperature was measured. The Mössbauer spectrum, which is shown in Figure 4b, shows the presence of iron(II). Therefore, it can be said that next to the polymer, the complex must be present. That the complex is neither oxidized nor decomposed through the electrospinning process can be assumed due to the measured isomeric shift (δ = 0.424 mm s^−1^) and the occurring small quadrupole splitting (ΔE_Q_ = 0.172 mm s^−1^) that can be observed in the spectrum, which is in the common range for iron(II) triazole complexes in the LS-state [14,31].

As the Mössbauer analysis showed us that iron was present in the composite, the UV/Vis measurements were performed to investigate the spin crossover properties as a color change occurs during the spin state switch from pink to white. Figure 4d shows a comparison of pure PMMA nanofibers with polymer complex composites (PCC) in the LS and HS state. In the range between 450 and 650 nm, a peak at 525 nm occurs at room temperature when the complex is in the low spin state according to the Mössbauer spectrum. This peak represents the significant band, as it is the only one in the spectrum of the visible light. As we increased the temperature, we observed a decrease in the absorption of said band. After reaching a certain temperature, no further decrease was observed. This has to be connected to the characteristic color change of iron(II)-triazole complexes from pink to white. Therefore, the spin crossover occurred during the heating process where the complex changes its spin state to HS.

To further analyze the switching behavior of the complex and the PCC, magnetic measurements with a SQUID magnetometer were performed. The measurements in Figure 5 also show that the SCO behavior was maintained with only a slight change of the occurring hysteresis. It was possible for the measurement of the pure complex [Fe(atrz)_3_]Cl_2_ to be illustrated by the molar magnetic susceptibility with a dependence on the applied temperature. For the case of PCC, the dependence of the magnetic moment of the composite in emu from the temperature is depicted, as it is also possible to depict the hysteresis in this way. The value for T_1/2_↓ of the complex could be found at ~342 K, and the T_1/2_↑ value could be found at ~350 K. The PCC, in comparison, showed a T_1/2_↓ value at ~345 K and a T_1/2_↑ value at ~353 K. Therefore, no notable change to the hysteresis occurred after the electrospinning process, but a slight shift of the switching temperature was observed.

Through the IR-Spectrum and the Mössbauer spectrum, we were able to show that the complex was present and not harmed after the electrospinning process, and the UV-Vis spectrum and the SQUID measurement showed that the SCO properties still remained. Through the SEM pictures and the EDX measurements, which are shown in Figure 6, we were also able to determine the position of the particles in the product which we obtained through the uniaxial electrospinning process. Through the SEM pictures, we could estimate that the complex was deposited on the surface of the obtained nanofibers as beading structures. The average nanofiber diameter was at around 450 nm as measured from the images. The average diameter of the positions of the beading structures was also determined and was 1.75 µm. Thus, it can be seen that the previously agglomerated particles fragmented into smaller particles during sonification. These positions had an average distance to each other of 14.2 µm. Using the EDX measurements, we investigated those beadings and could prove that the beadings were indeed the used complex. This is visible in Figure 6 on the right as the EDX mapping shows a punctual high concentration of iron, chlorine, and nitrogen at the position of said beadings, as the complex contains these, and the PMMA does not.

### 3.2. Coaxial Electruspun PMMA Nanofibers with [Fe(atrz)_3_](2ns)_2_

We obtained a pink-colored fiber mat using the coaxial electrospinning process like the uniaxial spun nanofiber composite. In Figure 7a, the IR-spectra of pure PMMA fibers, pure complex, and the coaxial spun composite are displayed.

As in this case, the composite contained more complex in the synthesis, and the intensity of the bands is more visible. Again, the C=O band at 1700 cm^−1^ is clearly assignable to the PMMA, and some of the most important bands of the complex are visible in the composite. This leads to the same conclusion that the complex is still present and did not succumb to the conditions of the electrospinning process.

A Mössbauer spectrum at room temperature was also, in this case, measured to prove that the obtained composite product contained the used complex and that the complex was not decomposed. The spectrum, which is shown in Figure 7b, indicates the presence of iron(II) in the LS-State with a typical isomeric shift (δ = 0.256 mm s^−1^) and quadrupole splitting (ΔE_Q_ = 0.361 mm s^−1^) for iron(II) triazole complexes in this spin state [14,31]. Therefore, it can be stated that this complex was also not oxidized or decomposed during the electrospinning process.

With the information gained through the Mössbauer analysis, it can be seen that iron(II) was present in the composite, and the UV/Vis measurements were performed to investigate the spin crossover properties as previously performed for the uniaxial spun composite. In Figure 7d, the comparison of the pure PMMA fibers and coaxial spun composite fibers at room temperature and high temperature is shown. Additionally, multiple UV/Vis spectra were performed for different temperatures, as we increased the potential of the Peltier element used. In these UV/Vis measurements, it was visible that a bond at 525 nm diminished with the increasing temperature. This behavior can be attributed to the SCO effect and the switching properties of the complex [Fe(atrz)_3_](2ns)_2_. To further analyze the switching properties of the PCC, the SQUID magnetometer measurements were performed. The SQUID measurement is depicted in Figure 8 and is compared with a SQUID measurement of the pure complex. The magnetic hysteresis curve of the PCC is shown by the magnetic moment in emu in dependence of the temperature. T_1/2_↓ for the PCC can be found at ~296 K and T_1/2_↑ at ~315 K. The hysteresis curve of the complex is depicted with the molar magnetic susceptibility in dependence on the temperature. For the complex, T_1/2_↓ can be found roughly at ~293 K and T_1/2_↑ ~317 K indicating a reduction in the hysteresis due to the implementation of the complex into the poly fiber and a slide shift of the SCO.

The SEM images provided information on the structure of the nanofibers. In Figure 7, it can be seen that the coaxial nanofibers were successfully made. Additionally, some single nanofibers are visible with a diameter under 100 nm, showing that the coaxial electrospinning process was completely ideal. Otherwise, an average fiber diameter of 320 nm could be determined for the produced composite fibers. Nevertheless, the overview shows that the great majority is coaxially spun. To determine the triazole-complex’s possible location, an EDX analysis was performed in the form of a visualized map.

The resulting graph in Figure 9 on the top right indicates that the presence of the [Fe(atrz)_3_](2ns)_2_ in the nanofiber is equally distributed. As the SEM picture on the left shows several nanofibers, the corresponding EDX map shows an exceeding number of signals of each element detected. Nitrogen and sulfur, which can be assigned to the complex, were evenly distributed in the fibers. Iron, which had to be part of the complex [Fe(atrz)_3_](2ns)_2_, could not be detected by EDX mapping, which can be explained by the low percentage of iron in the complex. Sulfur and nitrogen, which had to be present twice and four times as often due to the complex composition, could be detected better, respectively. The distribution of the complex in the fibers suggests that the previously agglomerated particles fragmented into smaller particles via sonification.

As visible through the SEM pictures, the obtained composites were different in regard to the distribution of the complex. The uniaxial spun fibers had beading structures on their fibers and the co-axial spun fibers had no directly visible agglomerations of complex on their surface. Therefore, the uniaxial spun composite is more likely to be affected by outer influences than the coaxial obtained product. This was furthermore investigated by applying a droplet of water on both products. The result of this experiment is shown in Figure 10. As expected, the PMMA fibers were not solved in the case of both composites. The containing iron(II) has a tendency to oxidize during extended water contact. Through this, it was more apparent to see that the complex of the fibers with the beading structures could be harmed by applying water, and the composites of the other attempt remained unharmed. The coaxial spun fibers had an outer shell of the same polymer material, which was detectable through the SEM pictures. This protected the complex from being rinsed away. Therefore, the location of the SCO particles has to be inside the produced fibers in comparison to the other obtained product.

## 4. Discussion

In a previous attempt, we also used amino-triazole complexes and gained composites with poly lactate acid (PLA). We choose SCO materials that show their switching behavior at low temperatures and analyzed the composites mainly using Mössbauer spectroscopy and the maintaining of the SCO properties after the electrospinning process. Therefore, the focus of our earlier study was not to determine the exact position of the used complexes [14]. We chose PMMA instead of PLA because the combination of triazole complexes and PMMA was found to be very efficient in another study in regard to the possible application in polymer-based optical waveguides [28]. In this study, we presented two successful methods of the implementation of iron(II)-triazole complexes into polymer nanofibers via (coaxial)-electrospinning. In both cases, the structural integrity of the iron-triazole complexes remained intact after the spinning process, which was indicated using spectroscopic methods. We were also able to maintain the SCO capabilities, proven by the temperature-dependent UV/Vis spectroscopy, the SQUID measurements, and the Mössbauer measurements. Since only a slight change was observed in the SCO temperatures of the composites compared with those of the complexes, no further chemical modification of the complexes would be required to further affect the temperature-induced SCO. Thus, the temperature range of the SCO effect is almost exclusively dependent on the complex used. The SEM pictures were used to determine the morphology of the composites and to show that the position of the complex was dependent on the chosen electrospinning method. Two different composites were obtained, which show minor differences in their occurring SCO effect due to the exchange of the anions of the complexes, which illustrates that this method configuration of the effect is applicable and has no further influence on the other properties of the composites. We were able to obtain a product in which the complex could be detected on the fibers as beading structures by uniaxial electrospinning. In the other attempt with coaxial electrospinning, we gained a product that had a homogeneous distribution of the complex in the fibers.

Although we used a coaxial setup to spin the PCC, we obtained a mixture of core-shaft-like nanofibers and an interpenetrating network (IPN). This is due to the usage of the same solvent for both solutions [28]. However, as there is a large number of interrelated parameters, it is difficult to predict a certain shape or structure. Therefore, small issues, for example, a formation of a droplet at the needle, during the coaxial electrospinning process, could have led to the simultaneous formation of IPN, core-shaft or bi-lobal fibers, as visible in Figure 9 [32,33,34]. Nevertheless, the experiment with the water droplet demonstrated that the SCO particles were not rinsed away when a coaxial setup was used. Due to this, the SCO complex had to be inside of the fiber structure.

As the location and reachability of the SCO in the composite was determined, a post-treatment of said SCO can be discussed. Although for the [Fe(atrz)_3_]Cl_2_ composite, the SCO is fairly accessible, it should be clear by the water droplet experiment that any wet chemical treatment would lead to a great loss of many SCO molecules. On the other hand, the [Fe(atrz)_3_](2ns)_2_ composite was not affected by the water. In this case, the issue lies with the accessibility of the SCO, as it is not feasible to modify that without destroying the fiber structure. Therefore, it would be reasonable to perform chemical modification during the complex synthesis if desired.

## 5. Conclusions

In this study, we synthesized and characterized two distinct PCCs using two different electrospinning methods. One of the PCCs exhibited beading complex structures on top of the obtained fibers, while the other was an IPN as observed by the SEM images. The position and the distance between the complex beadings were also identified and measured using SEM and EDX. Additionally, we conducted a water exposure test on the obtained products and observed a significant difference in their response. The supposed core–shell structures displayed no interaction with water, while the PCC with beading structures exhibited visible damage and rinsing of the complex. Importantly, both PCCs maintained their switching properties throughout the electrospinning process and remained unoxidized, which was confirmed by UV/Vis, IR, and Mössbauer spectroscopy as well as the SQUID measurements. Only minor differences were observed in the hysteresis of the SQUID measurements. This is an improvement in comparison to previous studies in which composite blends with PMMA had different properties than the used complexes [17]. This was the case for both composites, which leads to the assumption that the right parameters of the electrospinning process were found for PMMA-based PCCs. The spin crossover effect at around ambient temperature demonstrated the potential of electrospun fiber materials for future applications. In summary, our findings highlight the utility of electrospinning as a viable method to produce PCCs with well-preserved switching properties, paving the way for further research and exploration of these materials in a variety of applications. Furthermore, we utilized two distinct amino-triazole complexes with different corresponding anions, enabling us to adjust the spin crossover (SCO) temperature. This flexibility also allows for the solubility of these complexes to be improved, as certain anions were shown to increase solubility. For instance, the complex used with 2ns, which was first synthesized by Garcia et al., could be soluble in further attempts with other solvents [35]. The successful implementation of SCO materials into electrospun fibers could also lead to a simplified and advanced way of addressing the used complexes by external stimuli. Complexes that exhibit SCO behavior tend to be powders; therefore, fiber mats could be advantageous with their large and controllable surface. This could be of significance for sensor devices. For future applications in optical technology, a deep analysis of the optical properties of both fiber structures should be considered along with an assessment of whether they would be applicable in polymer-based optical waveguides. When the stimulus of the SCO would be energy-based, such as with light irradiation or temperature, protection against outer influences such as water (or other solvents) could be advantageous. Such gained advantages could also lead to possible applications in biological fields, e.g., in the human body.

## Figures and Tables

**Figure 1 polymers-15-02365-f001:**
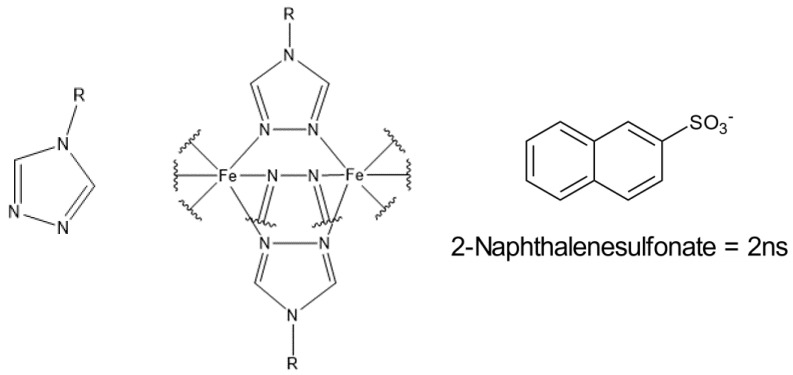
Triazole ligand (**left**) substituted at the 4-position, schematic illustration of the one-dimensional coordination chains (**middle**), and the 2ns anion (**right**). R refers to any organic rest at the 4-position.

**Figure 2 polymers-15-02365-f002:**
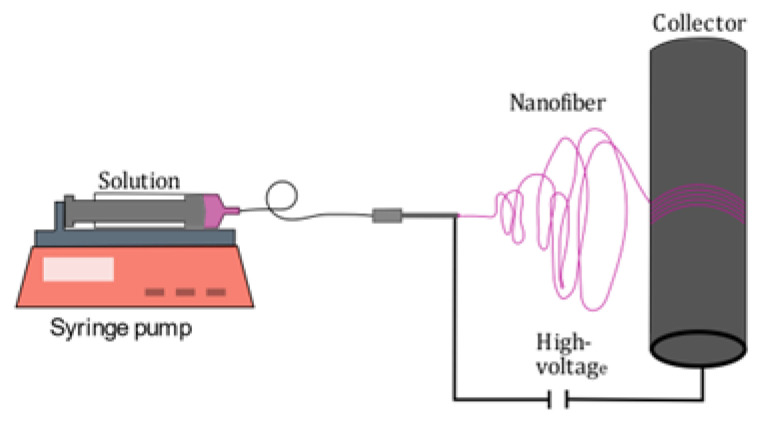
Deposition of [Fe(atrz)_3_]Cl_2_ onto PMMA nanofibers via electrospinning. During the emersion of the fiber jet, the solvent evaporates so that the solid polymer-complex nanofiber composite can be collected, using the rotating drum collector for further fiber alignment.

**Figure 3 polymers-15-02365-f003:**
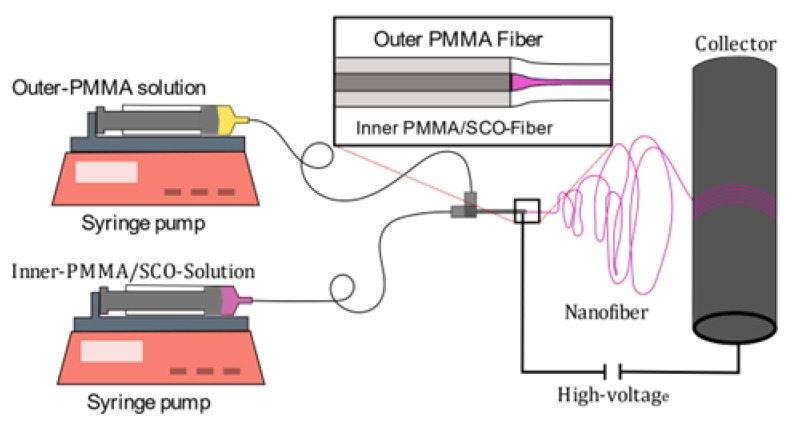
Implementation of [Fe(atrz)_3_](2ns)_2_ into PMMA nanofibers via coaxial electrospinning. The outer solution drags the inner solution with it to the collector, leading to the need of a higher pump rate of the outer solution. For fiber depletion, the same effect takes places as for the previous electrospinning process.

**Figure 4 polymers-15-02365-f004:**
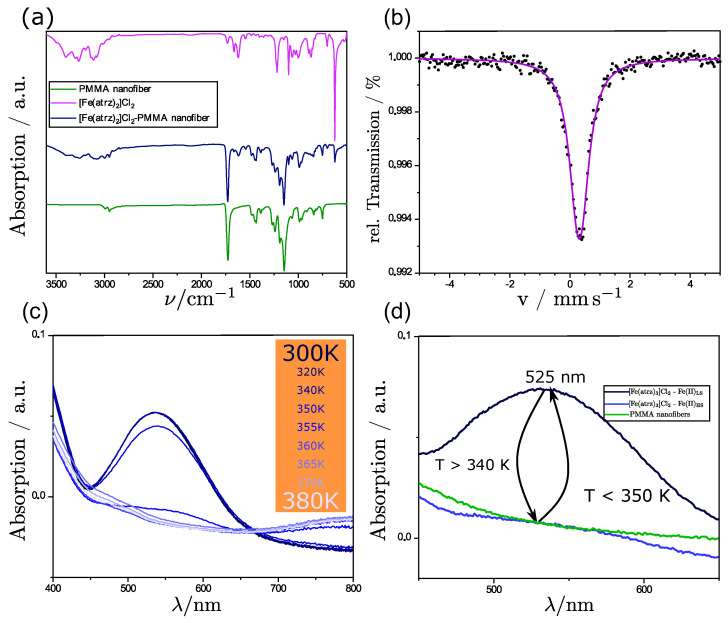
(**a**) Comparison of the IR-spectra of the pure PMMA fibers, the pure complex [Fe(atrz)_3_]Cl_2_, and the fiber complex composite. (**b**) Mössbauer spectrum of [Fe(atrz)_3_]Cl_2_ deposited on PMMA nano fibers at room temperature. (**c**) UV/Vis spectra of [Fe(atrz)_3_]Cl_2_ at different temperatures. (**d**) UV/Vis spectra around 525 nm of [Fe(atrz)_3_]Cl_2_ on the PMMA fiber at HS and LS state and pure PMMA fibers as comparison.

**Figure 5 polymers-15-02365-f005:**
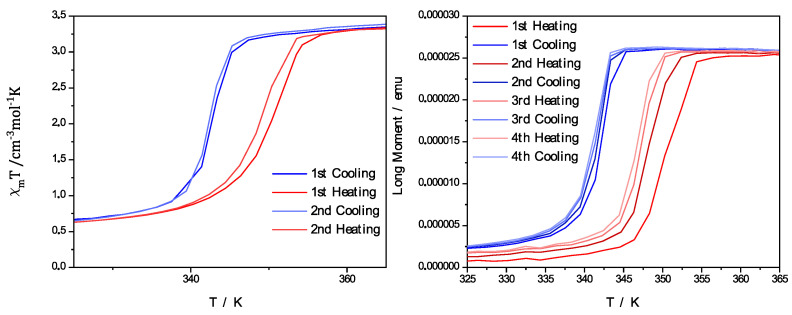
Magnetic measurements performed with a SQUID magnetometer. Pure complex [Fe(atrz)_3_]Cl_2_ shown on the left described by the magnetic susceptibility in dependence of the temperature. PCC on the right represented by magnetic moment in dependence of the temperature.

**Figure 6 polymers-15-02365-f006:**
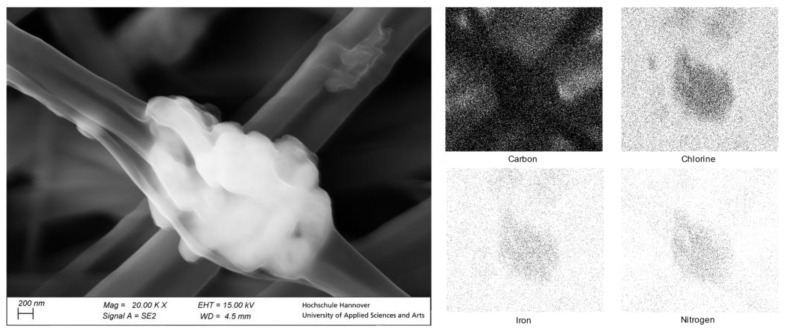
SEM Images of [Fe(atrz)_3_]Cl_2_ nanofiber composite. SEM-EDX measurement of [Fe(atrz)_3_]Cl_2_ on a mapping basis.

**Figure 7 polymers-15-02365-f007:**
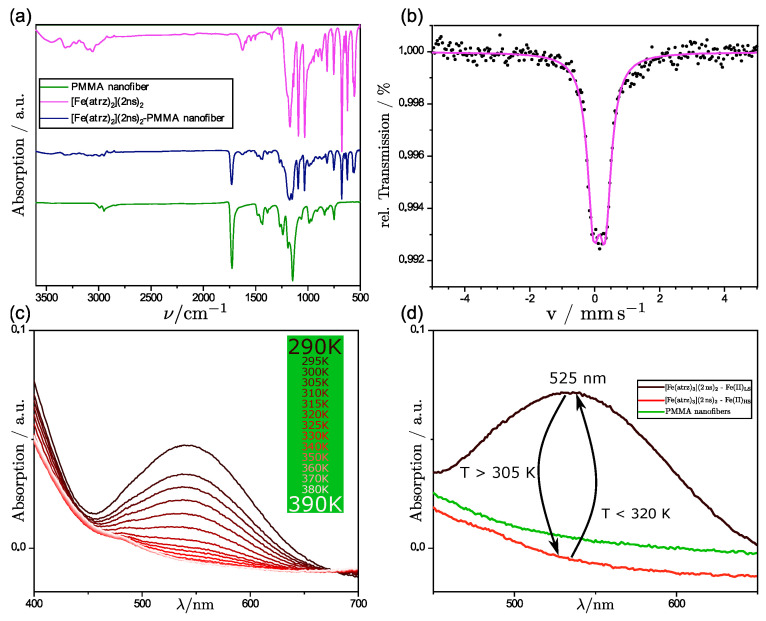
(**a**) Comparison of the IR-spectra of the pure PMMA fibers, the pure complex [Fe(atrz)_3_](2ns)_2_, and the fiber complex composite. (**b**) Mössbauer spectrum of the coaxial spun PCC nanofibers. (**c**) UV/Vis spectra of [Fe(atrz)_3_](2ns)_2_ at different temperatures. (**d**) UV/Vis spectra around 525 nm of PCC fiber at HS and LS state and pure PMMA fibers as comparison.

**Figure 8 polymers-15-02365-f008:**
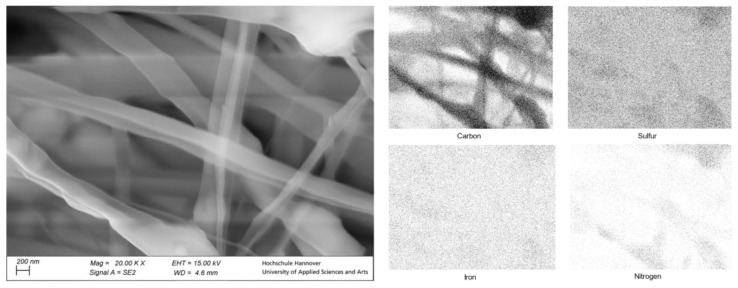
SEM of coaxial spun complex fiber composite with [Fe(atrz)_3_](2ns)_2_ as well as an EDX mapping.

**Figure 9 polymers-15-02365-f009:**
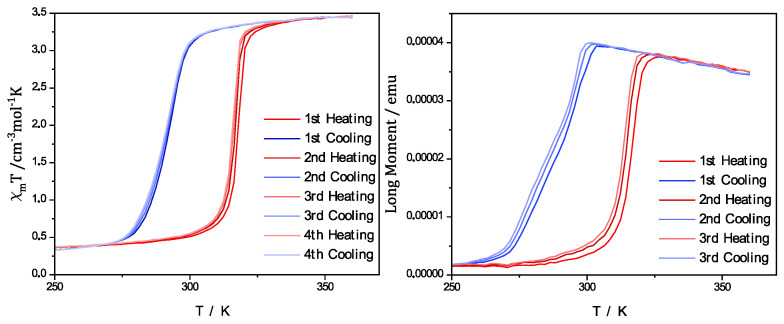
Magnetic measurements performed with a SQUID magnetometer. Pure complex [Fe(atrz)_3_](2ns)_2_ shown on the left described by the magnetic susceptibility in dependence of the temperature. PCC on the right is represented by magnetic moment in dependence of the temperature.

**Figure 10 polymers-15-02365-f010:**
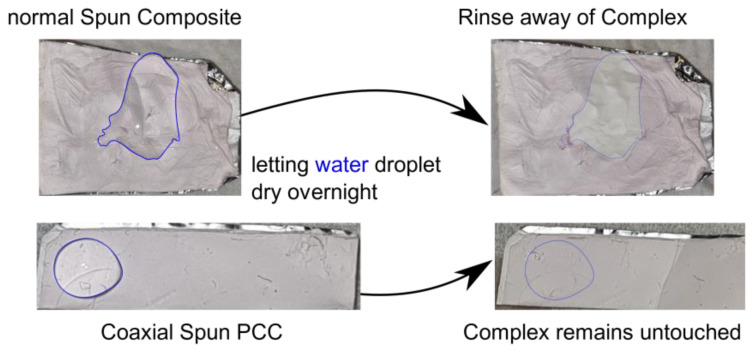
Difference of the effect of a water droplet cast on the nanofiber composite. Top is the regular electrospun composite, and bottom is the PCC where a coaxial nozzle was used. The blue line serves as visualization of where the water droplet is on the fibers.

## Data Availability

Link not yet provided.

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
