# Peer review of "Incorporation and Deposition of Spin Crossover Materials into and onto Electrospun Nanofibers"

_polymers, 2023, doi:10.3390/polym15102365_

Round 1
Reviewer 1 Report (Previous Reviewer 2)
Herein, the authors electrospin PMMA in which they incorporate an Iron-based spin cross over material. The authors note that the spin cross over properties are not affected by the PMMA medium. The authors synthesize their fibers using a typical nozzle, in addition one with inner and outer nozzles typically used for synthesizing core-sheath fibers. The study still suffers from a number of shortcomings:
1. What is the significance of incorporating the SCO material and observing no change in its properties? Are there other solid-state media that affect/inhibit SCO properties of this iron-based complex?
2. The authors do have provide evidence of a core-sheath structure; they are electrospinning composites.
3. Sentence punctuation should come after the citations, not before it.
4. Check all spacing and consistency of units (e.g., Line 107 >99% to >99 % and Line 168 is 1h, but on Line 169 it is 12 h)
5. What does unisized mean on Lines 178-179?
6. What does a cloth-like structure appear as?
7. Caption of Figure 6: amend the word composit to composite
8. Caption of Figure 8. Aswell amen to As well
9. Caption of Figure 10: Change up and down to top and bottom
10. Figure 10 belongs in the results section, not the discussion
11. Intuitively, the PMMA + SCO fibers with the single nozzle show via SEM that the SCO is towards the periphery of the fiber (as expected due to the intense whipping processing of electrospinning), hence, why the addition of water on the surface of the fibers dissolve some of the SCO. However, when the authors employ the dual nozzle, the additional surface layer of PMMA is able to better encapsulate the SCO during the electrospinning process so it can remain away from the periphery -- the additional PMMA can effectively 'shield' the SCO from the droplets of water. Can the authors make this idea more clear? They do note that in the former the SCO is on the periphery of the fiber but lack a clear and comprehensive discussion regarding this result.
English language is verbose and includes sections that read informally, which makes some of the sentence difficult to follow.
Author Response
Please see attachment

Reviewer 2 Report (New Reviewer)
The paper describe a nice use of electrospining to generate an SCO-decorated PMMA fiber system. I can consider acceptable the manuscript but I would really appreciate a bigger effort in describing potential applications of this new composite material. Beyond that temperature dependent mossbauer or raman experiments or other temperature dependent experiments performed on the final material should be provided as support to the reported SQUID investigation.
Round 2
Reviewer 1 Report (Previous Reviewer 2)
From the reviewers response, change clot-like to beading (the broad/general term the electrospinning field uses for this type of observation).
change composit to composite in caption of Fig 8.
The authors state:
"If there is no or only a slight change in the SCO
63 properties after implementation, no further chemical modification of the complexes is
64 necessary and the composite production can be simplified"
The authors just simply mix two things together and observe no changes with respect to SCO. Chemistry is not a matter of just juggling chemicals. The authors need to include more background with regard to how this work is significant and the results important. How it is very challenging to incorporate an SCO into a material AND keep virtually identical properties? They should do these including references or review articles. What was done previously? What were the limitations to those methods? Was it the material? Was it because the SCO was dissolved in solution? Is it because SCO is affected by thermal/UV/pressure, but not electrical and those conditions are not employed here? These are the questions the authors need to address. The addition the authors added above speaks nothing to this and creates more confusion -- can you modify the complexes after they are embedded in the fiber? Is 'no or a slight change' good enough?
Author Response
Please see the attachment.

This manuscript is a resubmission of an earlier submission. The following is a list of the peer review reports and author responses from that submission.
Round 1
Reviewer 1 Report
In the submitted manuscript, the authors deal with the synthesis of iron(II)-triazole spin crossover compounds and their incorporation into PMMA fibers using electrospinning. Research related to iron(II)-triazole spin crossover compounds is generally very interesting. However, this manuscript does not have a good concept and suffers from a number of shortcomings. This manuscript needs to be completely revised so that it can be published.
Here are the basic recommendations for major revisions:
Significantly improve English, both grammatically and stylistically, correct a large number of typos.
Introduction:
Lines 64-65, describing typical electrospinning as a setup where there is a rotating grounded collector is not accurate. The collector very often appears in the basic setup in a different form and polarized. Fiber jests is not the right turn, ect..
Lines 69-86, this text should rather be placed at the end of the introduction. The last part of the introduction should also contain a more clearly described goal of the manuscript, as well as the possibilities of future use that this system offers.
Are there other studies that describe the incorporation of these compounds into fibers using electrospinning? If not, it would be good to mention that too. What is the difference in the optical activity of PLA and PMMA polymer? How significantly is PMMA optically active than a common amorphous polymer in relation to these substances? It would be good to explain.
Materials and methods
at least a basic description of analytical methods such as all spectroscopy, magnetometry, SEM etc. is completely missing.
The description of the electrospinning process is completely inadequate, there are no needle diameters, polarity, temperature, humidity, distances etc...
It is completely missing why a different substance is used for classic needle electrospinning than for coaxial electrospinning
From the description of the synthesis, it follows that the product is a powder, in relation to the incorporation into nanofibers, therefore the particle size of this product is missing.
Figure captions are very special, this text should be transferred to appropriate chapters such as methodology and results.
The results
Figure 4a - the individual curves are not presented in the same intensity, visible for example on different noises ect.. if the intensities are compared, the significant peaks with which the authors demonstrate the presence of compounds will be very weak and the whole evidence very inconclusive.
SEM images need to be improved, despite the fact that the fibers contain a metal complex, it is possible to achieve much better quality SEM images.
According to the SEM images, it is clear that we cannot talk about nanofibers, but microfibers, maximally submicron fibers.
Despite the fact that the article is primarily focused on the properties of the incorporated compounds, there is no structural analysis of the fiber layers - homogeneity, fiber diameters. The description of these hooks, for example, incorporation, also affects the productivity of the process and the structural properties.
The description of the distribution of the compound in the fibers in both coaxial and uniaxial spinning is insufficiently supported by the results and methods. EDX is not carried out on a surface and map basis. Conjecture about a "surface-located compound" in uniaxial spinning is at least strange considering other articles dealing with the incorporation of inorganics into fibers using electrospinning.
Finding a better distribution of the compound using electrospinning is very debatable when a different complex was used than in the case of uniaxial spinning. SEm scales in Figure 9 are incomparable to Figure 6.
Discussion
This part is completely insufficient, it only summarizes the results presented in the previous part in a different form. The authors do not discuss or compare with the available literature, but only state what has been achieved.
It is inappropriate to refer to the discussion of manufactured fibers as optical fibers.
Conclusion
It is missing, nowhere is it stated what the results can lead to further, What are the perspectives in the examples or what are the other research plans that would further develop this study.
_____________
Overall, this article cannot be recommended for publication and needs a complete overhaul
Reviewer 2 Report
I have several point-by-point remarks, however, the manuscript has a significant amount of shortcomings so I will speak to them overall:
1) There is no experimental evidence of a core-sheath structure. Further, PMMA+dopant was used as a 'core' and PMMA was used as a 'sheath'. How do the authors know that the layers of PMMA in TFE did not just mix during the whipping stage of electrospinning? I'd expect them to given that the same solvent was used.
2) Materials and methods section is missing all instrumental information: UV/Vis, FTIR, Squid, SEM, and Mossbauer. How are you taking UV-Vis of a fiber? Is it MSP? Inner and outer diameter of core/sheath syringes are missing.
3) Virtually all figure captions (notably Figures 4 through 7) contain text that belongs in an results and discussion section, not a caption.
4) Conclusions section is missing
5) Several typos: 'quadrupl' should be 'quadrupole'; 'destinc' should be 'distinct'; among others.
6) Several areas of non-scientific adjectives are used: 'color-giving band'; 'overshadowed'; among others.
7) The authors must use the English style of numerical puncutation: 0,5 g should be 0.5 g. In some areas a space between the number and qualifier is missing: (e.g., 1h should be 1 h)
8) The text is too verbose in most areas and retracts from the main points and conclusion of the article.
Reviewer 3 Report
Maximilian Seydi Kilic et al. synthesized iron(II)-triazole-complexes, and further deposited on PMMA fibers or encapsulated into PMMA fibers, and finally explored their properties. Some minor revisions should be conducted before publication.
1. The Abstract section should be rewritten. The authors paid more attention on what they did. Instead, they should introduce more about their experimental results and conclusion. In addition, some important result data are recommended to be presented in this section.
2. Please state the merits and demerits of PMMA compared to the other commonly-used polymers biopolymers. In other words, why was PMMA chosen in this study?
3. Some more descriptions about electrospinning should be introduced in he introduction section. Some recent review works on the advances of electrospinning nanofibers like 10.1016/j.mtchem.2022.100944, and 10.3390/pr8060673, are suggested to be discussed.
4. In the Materials and Methods section, some important information are missing. For example, why were the currently-utilized electrospinning parameters chosen? Are there any preliminary experiments conducted? All the characterization methods with detailed testing parameters should be presented.
5. Figure 6: the inserted figure element is not clear. The figure with high quality should be provided. It also applied to Figure 9.
6. Some more proof should be provided to demonstrate the formation of coaxial nanofibers.
7. The grammar and writing should be improved in the whole manuscript.